# Prostaglandin E2 Exposure Disrupts E-Cadherin/Caveolin-1-Mediated Tumor Suppression to Favor Caveolin-1-Enhanced Migration, Invasion, and Metastasis in Melanoma Models

**DOI:** 10.3390/ijms242316947

**Published:** 2023-11-29

**Authors:** Lorena Lobos-González, Lorena Oróstica, Natalia Díaz-Valdivia, Victoria Rojas-Celis, America Campos, Eduardo Duran-Jara, Nicole Farfán, Lisette Leyton, Andrew F. G. Quest

**Affiliations:** 1Centro de Medicina Regenerativa, Facultad de Medicina-Clínica Alemana, Universidad del Desarrollo, Avenida Lo Plaza 680, Las Condes 7610658, Chile; llobos@udd.cl (L.L.-G.); edduran47@gmail.com (E.D.-J.); 2Advanced Center for Chronic Diseases (ACCDiS), Facultad de Ciencias Químicas y Farmacéuticas, Universidad de Chile, Santiago 8380494, Chile; nataliadiazvaldivia@gmail.com (N.D.-V.); victoria.rojcel@gmail.com (V.R.-C.); america.camposg@gmail.com (A.C.); 3Laboratory of Cellular Communication, Center for Studies on Exercise, Metabolism and Cancer (CEMC), Program of Cell and Molecular Biology, Biomedical Sciences Institute (ICBM), Faculty of Medicine, Universidad de Chile, Santiago 8380453, Chile; lorenaorostica@gmail.com; 4Centro de Investigación Biomédica, Facultad de Medicina, Universidad Diego Portales, Santiago 8370007, Chile; 5CRUK Scotland Institute, Glasgow G61 1BD, UK; 6Subdepartamento Genética Molecular, Instituto de Salud Pública de Chile, Santiago 7780050, Chile; 7Cancer and ncRNAs Laboratory, Universidad Andres Bello, Santiago 7550611, Chile; nicole.farfan@gmail.com

**Keywords:** Caveolin-1, E-cadherin, PGE2, inflammation, tumor progression

## Abstract

Caveolin-1 (CAV1) is a membrane-bound protein that suppresses tumor development yet also promotes metastasis. E-cadherin is important in CAV1-dependent tumor suppression and prevents CAV1-enhanced lung metastasis. Here, we used murine B16F10 and human A375 melanoma cells with low levels of endogenous CAV1 and E-cadherin to unravel how co-expression of E-cadherin modulates CAV1 function in vitro and in vivo in WT C57BL/6 or Rag−/− immunodeficient mice and how a pro-inflammatory environment generated by treating cells with prostaglandin E2 (PGE2) alters CAV1 function in the presence of E-cadherin. CAV1 expression augmented migration, invasion, and metastasis of melanoma cells, and these effects were abolished via transient co-expression of E-cadherin. Importantly, exposure of cells to PGE2 reverted the effects of E-cadherin expression and increased CAV1 phosphorylation on tyrosine-14 and metastasis. Moreover, PGE2 administration blocked the ability of the CAV1/E-cadherin complex to prevent tumor formation. Therefore, our results support the notion that PGE2 can override the tumor suppressor potential of the E-cadherin/CAV1 complex and that CAV1 released from the complex is phosphorylated on tyrosine-14 and promotes migration/invasion/metastasis. These observations provide direct evidence showing how a pro-inflammatory environment caused here via PGE2 administration can convert a potent tumor suppressor complex into a promoter of malignant cell behavior.

## 1. Introduction

Melanoma is the most aggressive type of skin cancer, and its incidence is increasing at a significant rate throughout the world. In this regard, melanoma cells, along with several types of skin cells (e.g., keratinocytes, sebocytes, and melanocytes), are characterized by the secretion of high levels of pro-inflammatory cytokines and prostaglandins (PG) in response to several pro-inflammatory stimuli [1,2,3]. For this reason, the progression of this particular type of cancer is strongly associated with chronic inflammation [1,2,3]. Among the many known PGs, PGE2 stands out as playing an important role in inflammatory processes. For instance, a number of studies have identified a positive correlation between chronic inflammation, detectable levels of PGE2, and intestinal carcinogenesis [4,5,6,7], and these observations have been confirmed in several animal studies [8,9,10,11]. PGE2 additionally promotes angiogenesis, tumor growth, and metastasis in breast cancer [12,13] and skin cancer [3,14,15,16]. However, despite such insight, the mechanisms by which PGE2 exposure favors tumor progression in different cancer models remain a matter of debate.

PGE2 is produced by a family of mammalian oxygenases known as cyclooxygenases. Specifically, cyclooxygenase-2 (COX-2), which is responsible for generating PGE2, has also been associated with inflammatory processes and disease [17]. For instance, COX-2 has been shown to be upregulated in a non-melanoma skin cancer (NMSC) model, where PGE2 levels have also been found to be increased. In this regard, an increased expression of COX-2 has also been observed in other types of cancer, such as colon cancer [17,18] which leads to increased levels of PGE2 being involved in ROS production, inflammation, and disease progression [19,20]. Moreover, in a pancreatic cancer model, COX-2 activity and PGE2 production via cancer-associated fibroblasts promote the motility of pancreatic tumor cells [21] and inhibit T-cell proliferation in a contact-independent manner [22].

PGE2 is known to increase the transcriptional activity of β-catenin-Tcf/Lef [23,24,25,26], as well as the growth [27,28] and viability of colon cancer cells [27,29]. In addition, the expression of COX-2 and the production of PGE2 stabilize survivin, an inhibitor of an apoptosis protein (IAP) family member that is overexpressed in many human tumors, including colon cancer [30], and increases resistance to apoptosis [31,32]. Furthermore, the expression of survivin, as well as COX-2, is directly regulated via the β-catenin/Tcf-Lef pathway [33,34,35]. More recent research has shown that melanoma cells proliferate [36] and migrate more when PGE2 receptor agonists are added [37], as opposed to when PGE2 receptor antagonists are present [37].

Specifically, migration of melanoma cells was induced after being treated with EP2 or EP4 agonists, suggesting that PGE2 might be a crucial new marker of melanoma progression [37].

We previously reported that the scaffolding protein Caveolin-1 (CAV1) exerts a role as a tumor suppressor in different types of cancer [38,39,40,41], and we linked this function to its ability to form a complex with E-cadherin (E-cad) and thereby facilitate the sequestration of β-catenin (β-cat) to the plasma membrane. In doing so, translocation of β-cat to the nucleus to initiate the transcription of genes such as survivin and VEGF is blocked [34,42,43,44]. Likewise, COX-2 expression, important for the production of pro-inflammatory PGE2, is subdued by the formation of the CAV1/E-cad complex [35]. However, we have also shown that in the presence of exogenous PGE2, the CAV1/E-cad/β-cat complex dissociates, releasing β-cat and enhancing the transcription of survivin and COX-2 [35]. 

Here, it is important to note that CAV1 not only behaves as a tumor suppressor but also as a promoter of metastasis. Indeed, our group has shown that in the absence of E-cad, CAV1 phosphorylation on tyrosine-14 (Y14) promotes signaling via a novel Rab5-Rac1 axis and, in doing so, enhances migration and invasion in vitro, as well as metastasis in a melanoma model in vivo [39,45,46,47,48]. With this in mind, we wondered whether exposure of E-cad/CAV1-expressing cells to PGE2 would suffice not only to promote transcriptional events dependent on β-cat release from the complex (as described previously), but also to release CAV1 from the complex with E-cad, increase CAV1 phosphorylation on Y14, and trigger CAV1-enhanced migration, invasion, and metastasis. In this study, we provide evidence that this is indeed the case.

## 2. Results

Our earlier work identified CAV1 as a potent negative regulator of genes whose expression favors the development and progression of cancer [39,44]. However, this was only the case in cell lines that co-expressed CAV1 and E-cadherin. In such cancer cells, CAV1 promotes β-cat recruitment to the plasma membrane, thereby precluding upregulation of β-cat/Tcf-Lef target genes like survivin and COX-2 [34]. COX-2 proteins are endoperoxide synthase enzymes that convert arachidonic acid into prostaglandin E2 (PGE2), among other prostanoids, initiating a sequence of events that interconvert various lipid mediators of inflammation [4,5]. Studies from our group have shown that treatment of cancer cells with PGE2 promotes the liberation of β-cat from the multiprotein complex that it forms with CAV1/E-cad [35]. Since, as indicated above, the COX-2 gene is a target of the β-cat/Tcf-Lef pathway, PGE2, a product of COX-2 activity, also regulates COX-2 expression. Therefore, a self-feeding amplification loop is generated, which may explain why the COX-2 genes are frequently upregulated in cancer and implicated in promoting metastasis. 

### 2.1. PGE2 Decreases CAV1/E-Cad Complex Formation and Increases the Phosphorylation of CAV1 on Tyrosine-14

B16F10 melanoma cells were stably transfected with either the empty vector (pLacIOP) or the plasmid with a CAV1-encoding insert (pLacIOP-CAV1). This plasmid allows IPTG-inducible expression of CAV1. Indeed, CAV1 levels increased in B16F10 (CAV1) cells in the presence of IPTG (Figure 1A,B) by around 10-fold compared to B16F10 (mock) cells (18 ± 2.0 and 1.7 ± 0.3, respectively; Figure 1B). E-cad was transiently expressed in B16F10 (mock) and B16F10 (CAV1) cells via transfection with pBATEM, a plasmid that encodes E-cad (8.8 ± 1.0 and 7.3 ± 0.9, respectively, Figure 1C). Following transfection, E-cad expression increased at least 7-fold in the B16F10 (E-cad) and B16F10 (CAV1/E-cad) cells when compared to non-transfected cells (Figure 1A,C). The formation of the CAV1/E-cad complex was evaluated in co-immunoprecipitation assays in the presence or absence of PGE2 in B16F10 (Mock), B16F10 (CAV1), B16F10 (E-cad), and B16F10 (CAV1/E-cad) cells (Figure 1D–G). Our group previously showed that CAV1 co-immunoprecipitates with β-cat in the presence of E-cad [44]. Consistent with previous findings, E-cad was readily detectable in CAV1 immunoprecipitates (Figure 1D) obtained from B16F10 (CAV1/E-cad) but not in B16F10 (CAV1) cells. Interestingly, in the presence of PGE2, the CAV1/E-cad ratio increased at least 3-fold compared to untreated B16F10 (CAV1/E-cad) cells (Figure 1E), indicative of a decrease in E-cad presence in these complexes. 

The same conditions were then employed to evaluate if increased phosphorylation of CAV1 on Y14 was associated with reduced CAV1/E-cad complex formation. As anticipated, after stimulating the cells with PGE2, and concomitant with the disruption of the CAV1/E-cad complex, Y14 phosphorylation increased at least 2-fold compared with the control (1.8 ± 0.05 and 0.9 ± 0.02, respectively; Figure 1F,G).

### 2.2. Exposure of B16F10 (CAV1/E-Cad) Cells to PGE2 Restores CAV1-Enhanced Migration and Invasion

To determine whether PGE2 treatment enhanced the migration and invasiveness of murine melanoma cells (Figure 2), B16F10 cells were cultured in the presence of 20 µM PGE2 for 24 h to then evaluate their migration and invasion potential. As previously reported, CAV1 expression in B16F10 (CAV1) cells increased their migration by at least 2-fold, while co-expression of CAV1 with E-cad reduced migration to levels seen for control cells (2.2 ± 0.2 and 1.0 ± 0.3, respectively; Figure 2A,B). The presence of PGE2 neither altered the migration of B16F10 (mock) nor B16F10 (CAV1) cells, but exposure to this pro-inflammatory prostaglandin enhanced the migration of B16F10 (CAV1/E-cad) cells to levels similar to those of B16F10 (CAV1) cells (1.9 ± 0.2 and 2.2 ± 0.4, respectively; Figure 2A,B). Essentially the same behavior was observed in the invasion assays. In B16F10 (CAV1/E-cad) cells, PGE2 exposure increased invasion approximately 3-fold, as compared with the same cells in the absence of this prostaglandin (3.3 ± 0.3 and 0.87 ± 0.8, respectively; Figure 2C,D). 

In summary, co-overexpression of CAV1 with E-cad suppressed CAV1-enhanced migration and invasion of B16F10 melanoma cells, and exposure of these cells to PGE2 abolished the ability of E-cad to restrict the pro-metastatic function of CAV1. 

### 2.3. Tumor Suppression Due to CAV1/E-Cad Co-Expression in B16F10 Cells Ablated via Exposure to PGE2

To test the effect of PGE2 on the tumor suppressor properties of the CAV1/E-cad complex in vivo, C57BL/6 mice were injected subcutaneously with B16F10 (mock), B16F10 (CAV1), B16F10 (E-cad), or B16F10 (CAV1/E-cad) cells in the absence or presence of PGE2 (−/+ PGE2), and the volume of the tumors was measured at different time points (Figure 3A). Mice injected with B16F10 (CAV1/E-cad) cells in the absence of PGE2 developed tumors with volumes of less than 200 mm^3^ by day 20 and of 203 ± 67 mm^3^ by day 24. In contrast, mice injected with B16F10 (CAV1/E-cad) cells treated with PGE2 exhibited tumors larger than 600 mm^3^ on day 20 and reached volumes of 1168 ± 183 mm^3^ by day 29. The difference between these two groups was even more remarkable on day 25, when B16F10 (CAV1/E-cad) cells in the absence of PGE2 formed tumors with a volume of only 193 ± 34 mm^3^ (Figure 3B.1–B.3). Of note, when B16F10 (CAV1) cells were injected in the presence or absence of PGE2, tumor volumes were 1363 ± 104 mm^3^ and 1022 ± 107 mm^3^, respectively, on day 20. The inhibition of tumor growth due to CAV1 presence was noticeable in comparison with B16F10 (mock) cells, for which tumor volumes were 1418 ± 235 mm^3^ on day 16 (Figure 3B.1–B.3). Thus, the ability of PGE2 to promote tumor growth was essentially limited to those conditions where B16F10 cells expressed both E-cad and CAV1.

To determine whether PGE2 exposure also modulated CAV1-induced metastatic capacity, B16F10 cells were injected into the tail vein of C57BL/6 mice (2 × 10^5^ per animal). After 21 days, mice were euthanized, lungs were collected and fixed, and tumor mass was evaluated (Figure 3C). As shown (Figure 3D), mice that were injected with B16F10 (CAV1/E-cad) cells in the absence of PGE2 showed fewer and smaller melanoma nodules (black nodules) on day 21. Total lung tumor mass was 0.16 ± 0.03 g, compared to mice injected with B16F10 (CAV1) or B16F10 (mock) cells, with values of 0.35 ± 0.03 g and 0.19 ± 0.01 g, respectively. Furthermore, the average lung weight in mice injected with B16F10 (CAV1/E-cad) in the absence of PGE2 was two times smaller than that of mice injected with B16F10 (CAV1). In these lungs, the metastatic nodules that developed had an approximate weight of 0.33 ± 0.02 g. Thus, the most remarkable change in these assays was the two-fold increase in metastasis (** *p* < 0.01) observed when B16F10 (CAV1/E-cad) cells were incubated with PGE2 prior to injecting the mice with B16F10 cells. 

When animals were euthanized, blood was collected from each mouse, and the serum levels of the pro-inflammatory cytokines IL-6, IL-10, IL-12, IFNγ, TNFα, and MCP1 were determined and found to be similar for all the animal groups (Appendix A). In addition, in these assays, body weight was not affected. Thus, the differences observed in tumor formation and metastasis were not attributable to systemic changes in the levels of pro-inflammatory cytokines.

### 2.4. Exposure to PGE2 Increases the Migration and Invasion of Human Melanoma Cells

We previously showed that the increased expression of CAV1 in A375 human melanoma cells, A375 (CAV1), treated for 48 h with IPTG, increased migration, invasion, and metastasis in comparison with A375 (mock) cells [39,40]. Here, we evaluated the effect of transiently transfecting these cell lines with pBATEM to overexpress E-cad in the presence or absence of PGE2 (−/+ PGE2) (50 μM) for 5 h. Thereafter, the cell extracts were prepared for immunoprecipitation experiments or cell migration and invasion assays (Figure 4). Following immunoprecipitation, CAV1 and co-immunoprecipitated E-cad were detected via immunoblotting (Figure 4A). For A375 (CAV1/E-cad) cells treated with PGE2, no differences in total CAV1 or E-cad levels were detected (Figure 4B,C). However, a 2.5-fold decrease in the E-cad that was co-immunoprecipitated with CAV1 was observed compared to the cells without treatment (Figure 4D). CAV1 phosphorylation on Y14 increased slightly when A375 shC and A375WT cells were cultured in the presence of PGE2 (Figure 4E).

In migration assays with A375 (CAV1) cells, PGE2 exposure led to a modest, insignificant decrease in the number of cells per field (64 ± 14), as compared to 77 ± 22 cells per field without PGE2 (Figure 4E). Transient expression of E-cad in A375 (E-cad) and A375 (CAV1/E-cad) cells reduced the number of transmigrated cells per field almost 2-fold compared with A375 (CAV1) cells in the absence of PGE2 (31 ± 4 and 23 ± 3 cells/per field, respectively; * *p* < 0.05). Interestingly, the migration of A375 (CAV1/E-cad) cells returned to levels similar to those of A373 (CAV1) cells when incubated with PGE2 (Figure 4E,F; * *p* < 0.05; 55 ± 3 cells/per field). 

In the invasion assays, treatment of CAV1-expressing A375 (CAV1) cells with PGE2 led to a modest decrease in the number of invading cells (170 ± 11 and 148 ± 9 cells/per field, respectively; Figure 4G,H). In the case of A375 (E-cad) cells, invasion increased noticeably in the presence of PGE2, from 61 ± 5 to 89 ± 4 cells/per field (** *p* < 0.01). For A375 (CAV1/E-cad) cells cultured in the absence of PGE2, the number of invading cells decreased at least ten times compared to A375 (CAV1) cells, namely to 14 ± 3 cells/per field. However, when A375 (CAV1/E-cad) cells were incubated with PGE2, invasion increased approximately seven-fold to 105 ± 10 cells/per field (*** *p* < 0.001), compared to the same cells without PGE2 treatment (Figure 4H). Therefore, PGE2 exposure enhances both migration and invasion of A375 (CAV1/E-cad) cells.

### 2.5. PGE2 Increases the Colony-Forming Capacity of A375 (CAV1/E-Cad) Cells

Colony formation in soft agar is considered a valid in vitro parameter to predict tumorigenic potential in vivo since a crucial characteristic of cancer cells is their ability to grow in an anchorage-independent manner [49]. To determine whether PGE2 affected this property in A375 cells, A375 (mock), A375 (CAV1), A375 (E-cad), and A375 (CAV1/E-cad) cells were seeded onto soft agar in 12-well plates (Figure 5), and the number of colonies over 100 μm in diameter was quantified after 3 weeks (Figure 5A). Expression of E-cad in A375 human cells reduced colony formation both in the absence and presence of PGE2 (3.5 ± 0.1 and 3.2 ± 0.1, respectively) (Figure 5B). These values were two-fold lower than those recorded for the A375 (mock) and A375 (CAV1) cells (namely 9.0 ± 0.2 and 8.0 ± 0.05, respectively). The change in colony-forming ability was even more significant when CAV1 and E-cad were co-expressed in A375 cells in the absence of PGE2 (1.00 ± 0.1 colonies/field), with a value 8-fold lower than that of A375 (CAV1) cells (Figure 5B). For A375 (CAV1/E-cad) cells exposed to PGE2, anchorage-independent growth increased 3-fold to 3.7 ± 0.1 colonies per field (n = 3; *** *p* < 0.001), suggesting that these cells recovered the ability to grow in an anchorage-independent manner in the presence of PGE2. 

### 2.6. PGE2 Increases the Metastatic Potential of Human Melanoma Cells

E-cad behaved like a classic tumor suppressor in tumor formation and metastasis assays when expressed in B16F10 (E-cad) cells, both in the absence and presence of PGE2 (Figure 3). Additionally, tumor suppression was enhanced when E-cad was co-expressed with CAV1 in B16F10 (CAV1/E-cad) cells (Figure 3B). 

To determine the relevance of this E-cad/CAV1 complex in a human model, we evaluated the metastatic potential of the A375 cells in the presence or absence of PGE2. Pathogen-free immunodeficient (C57BL6/Rag1−/−) mice between 5 and 7 weeks of age were used for these experiments. Specifically, A375 (Mock), A375 (CAV1), A375 (E-cad), or A375 (CAV1/E-cad) cells (2.5 × 10^6^ cells) pre-treated or not with PGE2 were injected intraperitoneally into these mice to then evaluate the metastatic potential of these cells (Figure 6A). In these experiments, tumor formation at any site is considered a reflection of the metastatic capacity of the cells injected.

In this model, tumors selectively develop in the spleen and the abdomen of the animals, and tumor growth is reported as an increase in the mass of the respective organs (tissue + tumor). Representative images of spleens with tumors are shown (Figure 6B). For mice injected with A375 (E-cad) cells, pre-treated or not with PGE2, both total spleen mass (0.32 ± 0.04 g and 0.33 ± 0.06 g, respectively; Figure 6C) and abdominal mass (0.29 ± 0.07 g and 0.15 ± 0.08, respectively; Figure 6D) were reduced, compared with values observed for mice injected with A375 (CAV1) cells (0.680 ± 0.003 g and 0.475 ± 0.005 g, respectively; Figure 6C,D). The most significant effect was observed in mice injected with A375 (CAV1/E-cad) cells, where total spleen mass and abdominal tumor mass (only solid tumor) were reduced compared to mice injected with A375 (CAV1) cells. This tumor suppressor effect was lost when A375 (CAV1/E-cad) cells were pre-treated with PGE2, and total spleen mass increased 2-fold to 0.458 ± 0.01 g (Figure 6C). The same behavior was observed for the abdominal tumor mass formed in the mice, where CAV1 promoted abdominal metastasis in A375 (CAV1) cells (0.475 ± 0.005 g), but E-cad partially decreased this behavior in A375 (E-cad) cells (0.286 ± 0.01 g).

Importantly, mice injected with A375 (CAV1/E-cad) cells hardly developed any tumor nodules in the abdominal cavity, with one exception (mean 0.008 ± 0.003 g), indicating that these cells are unable to undergo metastasis. However, when mice were injected with A375 (CAV1/E-cad) cells treated with PGE2, abdominal tumor mass increased 2-fold compared with the control group (0.458 ± 0.007 g total mass spleen and 0.168 ± 0.005 g abdominal tumor mass; *** *p* < 0.001, Figure 6C,D). 

In summary, exposure of both mouse and human melanoma cells expressing CAV1 together with E-cad overrides the tumor suppressor effect of this complex to enhance migration, invasion, and metastasis to levels comparable to those seen for cells expressing CAV1 in the absence of E-cad.

## 3. Discussion

CAV1 expression has reportedly been correlated with decreased presence of PGE2 in the culture medium of HEK293T and colon (HT29(ATCC) and DLD1) cancer cell lines [35]. This observation led us to evaluate the role of CAV1 when these cells were treated with PGE2. The results indicated that in the presence of this prostaglandin, CAV1, which forms a complex with E-cad [44], is no longer able to inhibit β-catenin-Tcf-/Lef-dependent transcription of genes associated with cell survival (COX-2 and survivin), angiogenesis (VEGF), invasion (MMP9), or metastasis (CD44) [43]. These data further suggested that PGE2 prevents stable formation of the CAV1/E-cad/β-catenin complex at the cell surface, which could lead to the CAV1-dependent development of malignant traits in vitro or in vivo in a preclinical model of metastasis [39]. Indeed, when we evaluated the formation of the CAV1/E-cad complex in the presence of PGE2 in murine melanoma B16F10 cells that simultaneously overexpressed CAV1 and E-cad, the CAV1/E-cad ratio in CAV1 immunoprecipitates decreased at least 3-fold when compared to untreated cells, suggesting that CAV1 was released from the complex with E-cad (Figure 1D–G). 

It is important to mention that simultaneous overexpression of CAV1 and E-cad in B16F10 cells not only decreases cell proliferation but also significantly enhances cell death [39,50]. Thus, considering the tumor-suppressor effects of E-cad and knowing that Y14 phosphorylation in CAV1 is associated with enhanced migration, invasion, and metastasis, our aim here was to evaluate whether CAV1 phosphorylation increased in the presence of PGE2 and promoted characteristics associated with malignant tumor cell behavior. As expected, PGE2 significantly increased the phosphorylation levels of Y14 when compared to untreated B16F10 cells (Figure 1F,G). These observations suggest that the formation of the CAV1 and E-cad complex prevents Y14 phosphorylation of CAV1 and, thereby, averts the pro-metastatic role of CAV1 [39,40,46]. 

We also corroborated these observations using the human melanoma line A375, which has elevated basal expression levels of CAV1 in comparison to B16F10 cells. There too, increasing the expression of E-cad via transient transfection with the plasmid pBATEM resulted in complex formation with CAV1, as indicated with co-immunoprecipitation experiments (Figure 4). Importantly, incubation of the A375 cells expressing both CAV1 and E-cad with PGE2 reduced the presence of E-cad in the co-immunoprecipitates with CAV1. Moreover, consistent with the results in B16F10 cells, phosphorylation of CAV1 tyrosine-14 in A375 cells was significantly increased in the presence of PGE2 (Appendix A). 

Given that the phosphorylation of CAV1 on Y14 is mediated with the non-receptor tyrosine kinases Src, Fyn, and Abl in response to a variety of stimuli [51,52,53], CAV1/E-cad complex disassembly with PGE2 might expose CAV1 to these kinases, thus increasing CAV1 phosphorylation. Indeed, PGE2 reportedly enhances c-Src activation and, therefore, promotes the migration of A549 lung cancer cells [54]. Although a role for CAV1 was not mentioned in this study, the observations may implicate CAV1 in this process since it is abundantly expressed in lung tissue [55]. Therefore, PGE2 may be envisaged as promoting migration via Src family kinase-mediated CAV1 Y14 phosphorylation. Alternatively, sequestration of a phosphatase in the CAV1-Ecad complex, which specifically reduces phosphorylation of CAV1, is also possible. Indeed, consistent with the latter possibility, we recently reported that in the presence of E-cad, dephosphorylation of CAV1 is promoted by recruitment of the tyrosine phosphatase PTPN14 [56]. Thus, additional studies are required to determine the precise mechanism by which PGE2 promotes CAV1 phosphorylation on Y14.

Enhanced phosphorylation on Y14 of CAV1 increases focal adhesion turnover, activation of Rac-1, and migration/invasion in metastatic melanoma, as well as breast cancer cells [48,53,57,58,59]. The opposite effect has been observed when E-cad is present; that is, E-cad cooperates with CAV1 to promote tumor suppression [39]. Therefore, CAV1 displays a dual role in cancer, not only according to the stage of tumor progression [50] but also depending on the presence or absence of specific proteins, like E-cad, which determine whether CAV1 will function as a tumor suppressor or tumor promoter [39,50,53]. What is still unclear is whether physiologically/pathologically relevant circumstances could induce a transition from tumor suppressor to promoter function. 

PGE2 is a pro-inflammatory prostaglandin in the context of cancer that has been shown to suppress the cytolytic activity of NK cells in thyroid cancer cells by inhibiting the expression of the natural cytotoxicity receptors [60] in order to promote the proliferation of osteosarcoma cells [61]. PGE1 also enhances migration and invasion in other types of cancer, such as hepatocellular carcinoma or non-small cell lung cancer, by upregulating the expression levels of Snail, a key inducer of epithelial–mesenchymal transition (EMT) [62,63], known to inhibit the expression of E-cad [64]. Here, we evaluated whether exposure to PGE2 sufficed to unleash the tumor-promoting role of CAV1, even in the presence of E-cad (Figure 3), and found that this was indeed the case, without the need for E-cad downregulation. Thus, together with the data available in the literature, we may posit that PGE2 promotes tumor development through both transcriptional and post-transcriptional mechanisms that eliminate E-cad function as a tumor suppressor.

The role of PGE2 has been examined in several in vivo cancer models. For example, exposure to this inflammatory prostaglandin leads to increased intestinal adenoma formation in F344 rats [65]. PGE2 also promotes epithelial cell proliferation and COX-2 expression via activation of the Ras-MAPK signaling pathway in mice, which spontaneously develop intestinal polyps and tumors [66,67,68,69,70]. These results are consistent with the possibility that PGE2 itself may promote the expression of COX-2, giving rise to a positive feedback loop that further increases PGE2 levels and thereby promotes the expression of mesenchymal stem cell markers. However, such effects are more likely to be observed as the consequence of systemic PGE2 elevations. Here, it should be noted that the melanoma cells were pre-treated with PGE2 before injection in the in vivo assays, making systemic effects unlikely. In addition, we measured systemic cytokine levels in blood samples from C57BL/6 mice injected with the different B16F10 cell populations and saw no significant alterations in the levels of the different cytokines (see Appendix A, (A) IL-10, (B) TNFα, (C) IFNγ, (D) IL-6, (E) IL-12, and (F) IL4), again suggesting that the results obtained were not attributable to systemic alterations in the recipient mice. Our previous studies showed that PGE2 can promote COX-2 expression and thereby enhance its own production even in the presence of CAV1/E-cad in colon cancer cells [35]. This generates a feed-forward amplification loop. Thus, it was interesting to determine whether something similar could happen in melanoma cells, where the CAV1/E-cad complex has a strong tumor suppressor effect [39]. Our results revealed that pre-incubation of B16F10 (CAV1/E-cad) cells with PGE2 led to enhanced subcutaneous tumor formation (Figure 3B) and increased lung metastasis in syngeneic C57BL/6 mice, compared to untreated B16F10 (CAV1/E-cad) mice (Figure 3D). 

Importantly, similar effects were observed using A375 human melanoma cells. First, the colony-forming ability of these cells CAV1/E-cad-expressing cells was substantially increased in the presence of PGE2 (see Figure 5). Furthermore, in intraperitoneal carcinomatosis assays in mice, the accumulation of tumor mass associated with the spleen and the walls of the peritoneal cavity are considered to reflect the metastatic potential. Here too, we observed a significant decrease in this ability for cells expressing CAV1/E-cad that was then reverted if the cells were pre-treated with PGE2 prior to injection. Finally, our results show that PGE2 exposure disrupts the tumor and metastasis suppressor effects of the CAV1/E-cad complex to favor CAV1 phosphorylation on tyrosine-14, resulting in enhanced migration, invasion, and metastasis of melanoma cells, as indicated in our working model (Figure 7).

CAV1, depicted here as a dimer associated with the plasma membrane, is shown to transit essentially between the free state and presence in a complex together with E-cadherin and β-catenin. The formation of the first complex is associated with the ability of CAV1 to inhibit β-catenin/Tcf-Lef-dependent transcription of genes including survivin, COX-2, and cyclinD1 and therefore function as a tumor suppressor. In the presence of PGE2, CAV1 dissociates from the complex and is phosphorylated on tyrosine-14, which we have previously shown to favor cell migration and invasion in vitro, as well as metastasis in vivo. Thus, PGE2 exposure to cells overrides the tumor suppressor capacity of the CAV1/E-cad complex and releases CAV1 to promote metastasis.

## 4. Materials and Methods

### 4.1. Materials

Rabbit polyclonal anti-CAV1 (ab18199 rabbit polyclonal), mouse monoclonal anti-CAV1 (C13620), mouse monoclonal anti-pY14-CAV1 (C611339) and mouse monoclonal anti-E-cad (C61018) antibodies were obtained from BD Transduction Laboratories, NJ, USA; monoclonal anti-β-actin antibodies (sc47778) were obtained from Santa Cruz Biotechnology (Dallas, TX, USA), and all were used as indicated by the manufacturers. Goat anti-rabbit and goat anti-mouse IgG antibodies coupled to horseradish peroxidase (HRP) were obtained from Bio Rad (Hercules Laboratories, San Diego CA, USA). The ECL chemiluminescence substrate and BCA protein determination kit were obtained from Pierce (Rockford, IL, USA). The Plasmid Midi Kit was from Qiagen (Valencia, CA, USA), and human fibronectin was from Becton Dickinson (San Jose, CA, USA). Hygromycin was obtained from Calbiochem (La Jolla, CA, USA), Isopropyl β-D-1-thiogalactopyranoside (IPTG) was from Sigma (St. Louis, MO, USA), and fetal bovine serum (FBS) was from Hyclone (Logan, UT, USA). Cell culture media and antibiotics were from GIBCO (Invitrogen, Carlsbad, CA, USA) and Fugene 6^®^ from Roche Applied Science (Roche, Basel, Switzerland).

### 4.2. Cell Culture

B16F10 metastatic murine melanoma cells (ATCC, #CRL6475, provided by Laurence Zitvogel, Institut Gustav Roussy, Villejuif, France) and the human melanoma cell line A375M (ATCC, #CRL1619), also employed previously [39], were maintained in RPMI 1640 medium. All media were additionally supplemented with 10% FBS, 100 U/mL of penicillin, and 100 μg/mL of streptomycin sulfate. Cells were cultured at 37 °C in a humidified atmosphere containing 5% CO_2_. B16F10 and A375M cell lines were stably transfected with the pLacIOP (referred to as mock) or pLacIOP-Caveolin-1 plasmids (referred to as CAV1, containing the full-length Caveolin-1 sequence, NCBI Reference Sequence: NP_001003296.1). Stable cell lines were selected and maintained in culture medium containing 750 μg/mL of Hygromycin. For the expression of E-cad, cells were transiently transfected with the pBATEM2 plasmid lacking a selection marker and encoding murine E-cad under the control of an actin promoter (provided by Amparo Cano, Universidad Autónoma de Madrid, Madrid, Spain).

PGE2 exposure: B16F10 (Mock), B16F10 (CAV1), B16F10 (Mock/E-cad), and B16F10 (CAV1/E-cad) cell lines were serum-deprived for 5 h before exposure to PGE2 (20 μM) for 24 h. A375M (Mock), A375M (CAV1), A375M (Mock/E-cad), and A375M (CAV1/E-cad) cell lines were treated with PGE2 (50 μM) for 6 h in the absence of serum. 

### 4.3. Western Blotting

Cells were rinsed and harvested in ice-cold PBS containing 1 mM of orthovanadate, 10 µg/mL of benzamidine, 2 µg/mL of antipain, 1 µg/mL of leupeptin, and 1 mM of phenylmethyl-sulphonylfluoride (Ova-BAL-PMSF). Cells were then centrifuged at 3000× *g* for 2 min at 4 °C, and the respective cell pellets were lysed via sonication in extraction buffer (20 mM Hepes pH 7.4, 0.1% NP40, and 0.1% SDS plus Ova-BAL-PMSF). Protein concentrations in extracts were determined using the BCA protein assay kit. Protein samples were separated using SDS-PAGE (50 μg per lane), transferred to nitrocellulose membranes, blocked in PBS containing 5% non-fat milk, and probed overnight at 4 °C with anti-CAV1 (1:5000) or anti-pY14-CAV1 (1:3000) antibodies diluted in PBS containing 10% gelatin and 1% Tween-20. Equal protein loading in each lane was confirmed by detecting β-actin [anti-β-actin antibody (1:5000)] as a loading control. Goat anti-mouse and anti-rabbit IgG antibodies coupled to horseradish peroxidase were used to detect bound first antibodies using EZ-ECL. Protein bands were quantified with a densitometric analysis using the Rasband, W.S., ImageJ, U.S. National Institutes of Health, Bethesda, MD, USA, https://imagej.nih.gov/ij/, 1997–2018 (available from NIH at accessed on 20 July 2023 https://rsb.info.nih.gov/).

### 4.4. CAV1 Immunoprecipitation

CAV1 immunoprecipitation was performed using protein A/G coupled to sepharose beads (Santa Cruz Biotechnology), according to the manufacturer’s indications; 2.5 μg of polyclonal anti-Caveolin-1 (ab18199 rabbit polyclonal) antibody diluted in 200 μL of PBS-0.1% Tween were incubated with 50 μL of protein A/G-sepharose for 2 h at 4 °C on a rotating shaker. Then, 2 mg of proteins in 500 μL of PBS-0.1% Tween were added, and the incubation was continued for another 12 h at 4 °C. Sepharose beads were separated via centrifugation and washed 3 times with PBS, and then 70 μL of loading buffer were added to solubilize samples for Western blotting analysis.

### 4.5. Migration and Invasion Assays 

Briefly, the inserts were coated on the lower surface with 2 µg/mL of fibronectin as a chemoattractant. Cells (1.5 × 10^5^) re-suspended in serum-free medium were added to the upper chambers, and serum-free medium was added to the bottom chambers. After 16 h, inserts were removed and washed, and cells that migrated to the lower side of the inserts were stained with crystal violet in 2% ethanol and counted in an inverted microscope.

Cell migration was evaluated in Boyden Chamber assays (Transwell Costar, 6.5 mm diameter, 8 µm pore size), and invasion was evaluated in Matrigel assays (BD Biosciences, 354,480), as previously reported [45,46].

Clonogenic assays: A375M (Mock), A375M (CAV1), A375M (Mock/E-cad), and A375M (CAV1/E-cad) cells were treated for 5 h with PGE2 (50 μM). Then, these cells were seeded (3 × 10^3^ cells) in 0.2% agarose in low-adhesion plates. After 24 h, medium supplemented with 10% FBS and PGE2 was added to each well as indicated, and 7 days later, the colonies larger than 50 μm were considered viable. On day 14, colonies were quantified, and only colonies larger than 100 μm were considered for further analysis, as previously described [71,72].

### 4.6. Animal Studies

Animal studies were conducted in accordance with the appropriate guidelines of the Ethical Committee of the Fundación Ciencia & Vida. Immune-compromised NOD/SCID mice were obtained from Jackson Laboratories (Bar Harbor, ME, USA) and maintained in the animal facility of the Fundación Ciencia & Vida under specific pathogen-free conditions (Tecniplast SpA, Buguggiate, Italy), in a temperature-controlled environment with a 12/12 h light/dark schedule and sterile food and water ad libitum. In the results section, an IC of 95%, a value of significance of 5% or less (*p* < 0.05), and a power calculation of 80% were accepted as statistically significant. All statistical tests were performed using GraphPad Prism 6 software (GraphPad Prism, Boston, MA, USA). An r package known as pwr2 was used to calculate the minimum sample size. The “ss.1way” test was used for a one-way balanced ANOVA model (k: number of groups; alpha: level of significance; probability of type I error, beta: probability of type II error (power = 1 − beta); f: effect size; delta: indication of the smallest difference between groups; k, sigma: indication of the standard deviation; and B: number of iterations, where the default value is 100).

### 4.7. Tumor Growth Assays

Mice between 6 and 8 weeks of age were used for experiments (approved by the local bioethics of FCV). B16F10 (Mock) (n = 8/8), B16F10 (CAV1) (n = 8/8), B16F10 (E-cad) (n = 9/9), and B16F10 (CAV1/E-cad) (n = 10/8) cells (1 × 10^5^ cells), either pre-stimulated or not, with 50 µM of PGE2 for 16 h (n = mice injected with pre-stimulated cells/mice injected with non-stimulated cells), were injected subcutaneously into the flanks of C57BL/6 mice. Tumor growth was monitored via palpitation (volume = width2 × length × π/6), as described [39].

Metastasis assays with B16F10 murine melanoma cells: immune-competent C57BL/6 mice were injected intravenously with 2 × 10^5^ B16F10 (Mock), B16F10 (CAV1), B16F10 (E-cad), and B16F10 (CAV1/E-cad) cells, pre-stimulated or not with 50 µM of PGE2 for 16 h. Each group included eight mice. Then, the animals were euthanized on day 21 post-injection, and their lungs were fixed in Fekete’s solution. Black tissue, corresponding to lung metastases, was evaluated by comparing it to total lung mass (g), as previously described [39,40,44,71]. 

### 4.8. Metastasis Assays with Human A375 Melanoma Cells

Pathogen-free immunodeficient B16Rag1−/− (B6.129S7 Rag1<tm1Mom>/J) mice were obtained from Jackson laboratories and maintained as indicated above. Immunosuppressed C57/B6Rag1−/− mice were inoculated via intraperitoneal injection with 2.5 × 10^6^/250 µL of A375M (Mock), A375M (CAV1), A375M (E-cad), and A375M (CAV1/E-cad) cells, pre-stimulated or not, with 50 µM of PGE2 for 5 h in serum-free medium. Then, spleen and abdominal metastasis were quantified using parameters such as the total mass of spleens with tumors and abdominal tumor mass.

### 4.9. Cytokine ELISA 

Serum levels of interleukin-6 (IL-6), interleukin-10 (IL-10), interleukin-12 (IL-12), interferon-γ (IFN-γ), tumor necrosis factor (TNF), and interleukin-4 (IL-4) of mice that were previously injected subcutaneously with B16F10 (Mock), B16F10 (CAV1), B16F10 (E-cad), and B16F10 (CAV1/E-cad) cells, pre-stimulated or not with 50 µM of PGE2 for 16 h, were determined using sandwich ELISA (BD Biosciences, San Diego, CA, USA), according to the manufacturer’s instructions.

### 4.10. Statistical Analysis 

All data are expressed as the mean ± standard error of the mean (SEM) of three independent experiments. Data were analyzed using the non-parametric Kruskal–Wallis test for multiple comparisons, with a Dunn post-test. Significance (*p*-value) was set at the nominal level of *p* < 0.05. All data were processed using INSTAT v 3.05 (GraphPad Software, San Diego, CA, USA, http://www.graphpad.com accessed on 20 July 2023).

## 5. Conclusions

According to our previous results, CAV1 and E-cad are present at the plasma membrane, forming CAV1/E-cad complexes that sequester β-catenin, thereby preventing its translocation to the nucleus and β-catenin/Tcf-Lef-dependent transcription of genes, including survivin and COX-2 [35,44]. Consistent with results from this study, pro-inflammatory PGE2 is shown to dismantle the CAV1/E-cad complex and release CAV1, which results in elevated phosphorylation on Y14, increased migration, invasion, and metastasis. By dissociating the complex, PGE2 also promotes β-catenin-dependent COX-2 expression to further increase PGE2 levels (previous studies). Thus, PGE2 dissociates the CAV1/E-cad tumor suppressor complex to promote its own production (positive feedback loop) as well as metastasis via CAV1 release and enhanced Y14 phosphorylation.

## Figures and Tables

**Figure 1 ijms-24-16947-f001:**
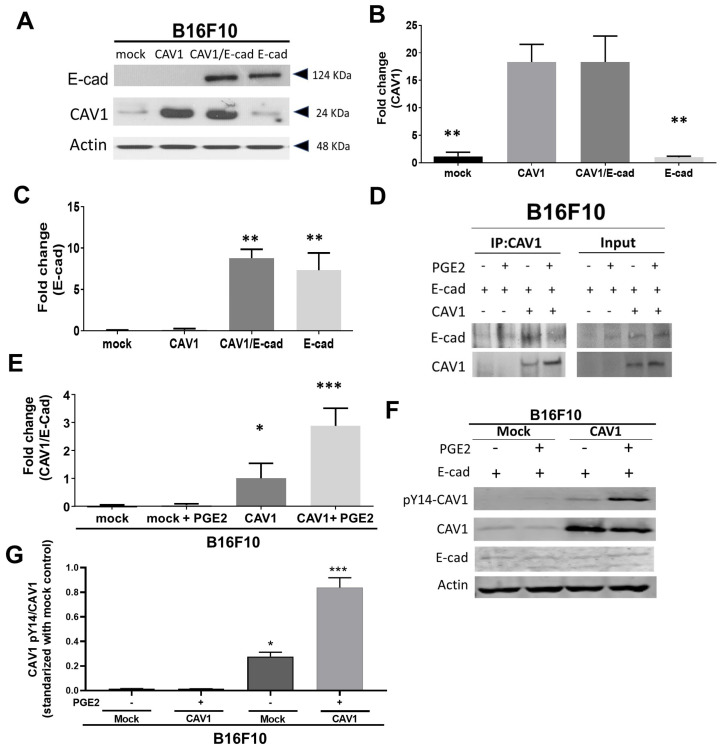
PGE2 reduces CAV1/E-cad complex formation and augments phosphorylation of CAV1 on tyrosine-14 in B16F10 cells. B16F10 (mock) and B16F10 (CAV1) cells were cultured in the presence of IPTG (1 mM) and then transiently transfected (or not) with pBATEM to overexpress E-cad. After 48 h, cell extracts from B16F10 (mock), B16F10 (CAV1), B16F10 (mock/E-cad), and B16F10 (CAV1/E-cad) cells were separated using SDS-PAGE in 12% minigels (50 μg total protein per lane). CAV1 and E-cad levels were evaluated via Western blotting using specific antibodies. Actin was used as a loading control. (**A**) Results from a representative immunoblot experiment. CAV1 (**B**) and E-cad (**C**) protein levels were quantified in several experiments via scanning densitometry and normalized to actin. (**D**) B16F10 (mock/E-cad) or B16F10 (CAV1/E-cad) cells were serum deprived for 5 h and treated or not with PGE2 (20 μM) for 2 h. Then, CAV1 was immunoprecipitated from protein extracts using a polyclonal antibody. Immunoprecipitated CAV1 and co-immunoprecipitated E-cad were detected via immunoblotting. The left panel shows a representative result, and the blot on the right shows the total protein input of each sample. (**E**) The values in the graph show the CAV1/E-cad ratios normalized to actin. (**F**) Protein extracts from B16F10 (mock) and B16F10 (CAV1) cells treated with 20 μM PGE2 or vehicle (control) for 2 h. Phospho-CAV1 (pY14-CAV1) and CAV1 and E-cad levels were evaluated via immunoblotting. Results from a representative experiment are shown. (**G**) pY14-CAV1, CAV1, and E-cad levels were quantified via scanning densitometry and are shown as the pY14-CAV1/CAV1 ratio standardized to mock (−) controls. Values in all graphs are the mean ± SD, n = 3; * *p* < 0.05, ** *p* < 0.01, and *** *p* < 0.001.

**Figure 2 ijms-24-16947-f002:**
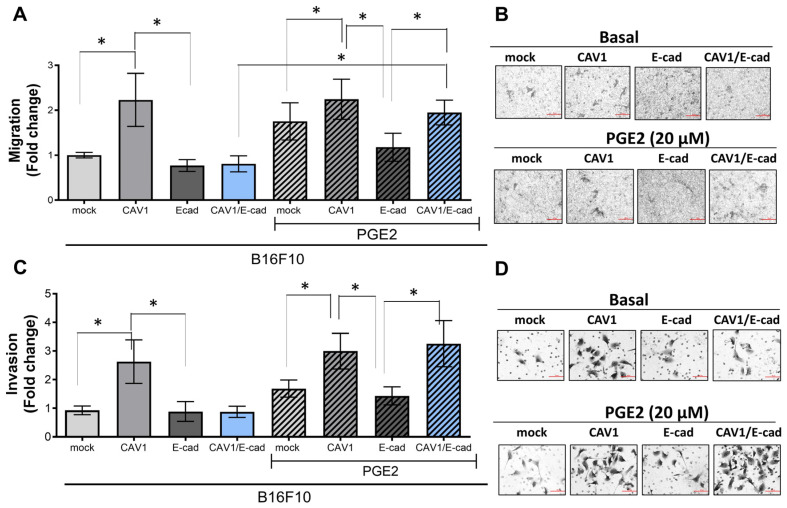
PGE2 enhances migration and invasion of B16F10 (CAV1/E-cad) cells. B16F10 (mock) and B16F10 (CAV1) cells cultured in the presence of IPTG (1 mM) were transiently transfected with pBATEM to overexpress E-cad. After 48 h, B16F10 (mock), B16F10 (CAV1), B16F10 (E-cad), and B16F10 (CAV1/E-cad) cultures were treated with 20 μM PGE2 for another 24 h, and then cell migration and invasion were evaluated. (**A**) B16F10 (mock), B16F10 (CAV1), B16F10 (E-cad), or B16F10 (CAV1/E-cad) cells (150,000) treated or not with PGE2 were seeded in Boyden chambers (transwells) in the presence or absence of 20 μM PGE2 and allowed to transmigrate for 16 h. The graph shows the fold change in the migration of cells treated or not with PGE2, normalized to B16F10 (mock) control cells. Significant differences are shown (* *p* < 0.05, n = 3). (**B**) The panels show representative images from the migration assays of cells in the absence (**upper panel**) or presence (**lower panel**) of PGE2. Magnification bar = 50 µm. (**C**) Cells treated or not with PGE2 (20 µM) for 24 h were used for the invasion assays. Cells (150,000) were seeded in matrigel Boyden chambers in the absence or presence of PGE2 (20 μM). Cell invasion was measured after 24 h. The graph shows the fold change in invasion normalized to B16F10 (mock) control cells. Significant differences are shown (* *p* < 0.05; n = 3). (**D**) The panel shows representative images of the invasion assays in the absence (**upper panel**) or presence (**lower panel**) of PGE2. Magnification bar = 50 μm.

**Figure 3 ijms-24-16947-f003:**
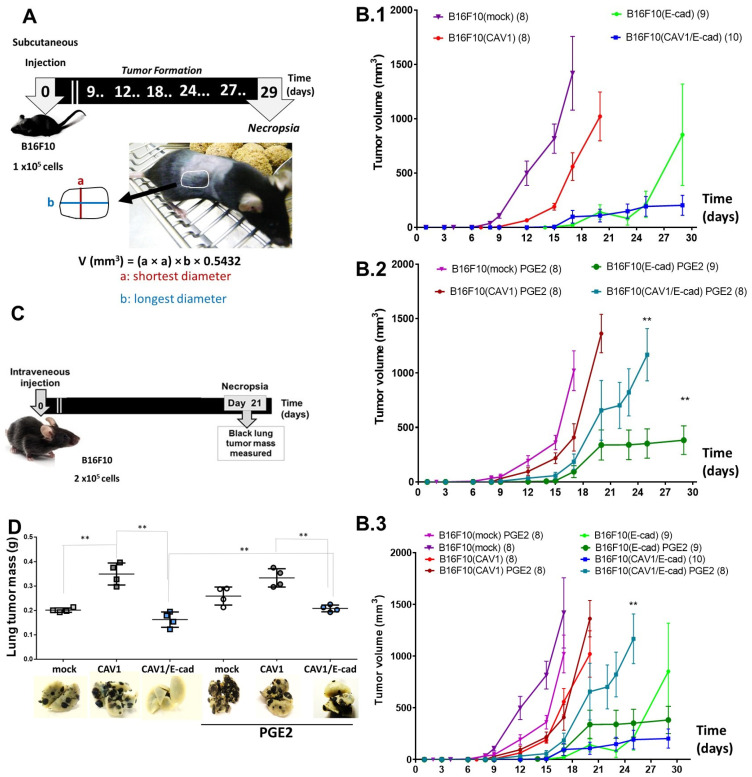
PGE2 enhances subcutaneous tumor formation and lung metastasis of B16F10 (CAV1/E-cad) cells. (**A**) Scheme illustrating experimental procedures in tumor formation assays. C57BL/6 mice between 5 and 7 weeks of age were used for these assays. Mice were injected subcutaneously (day 0) with B16F10 (mock), B16F10 (CAV1), B16F10 (E-cad), or B16F10 (CAV1/E-cad) cells (1 × 10^5^), pre-treated or not with 20 μM of PGE2 (for 24 h). At different time intervals, up to 29 days, tumor size was evaluated in all experimental groups, as indicated in the formula. (**B.1**–**B.3**) Graph showing the tumor volume (mm^3^) for animals at different time points post-injection of cells pre-treated or not with PGE2, and the cells injected were B16F10 (mock), B16F10 (CAV1), B16F10 (E-cad), or B16F10 (CAV1/E-cad). The results are shown as the mean ± SEM (n = 3; ** *p* < 0.01). (**B.1**) Graph showing a time-course experiment for mice injected with cells that were not previously incubated with PGE2. (**B.2**) Graph of time-course experiment for mice injected with cells that were previously incubated with PGE2. (**B.3**) Graph showing the curves for all time-course experiments. The details concerning the number of mice employed are described in the methodology section. (**C**) A scheme illustrating the experimental procedures in metastasis assays. Cells in culture pre-treated or not with 20 μM of PGE2 for 24 h were harvested, re-suspended in physiological serum, and then mice were injected intravenously in the tail vein with 2 × 10^5^ of B16F10 (mock), B16F10 (CAV1), B16F10 (E-cad), or B16F10 (CAV1/E-cad) cells (2 × 10^5^). After 21 days, the mice were euthanized and necropsied. The black lung tumor mass was quantified. (**D**) The graph shows lung tumor mass (g) in different experimental animal groups, and representative images of tumor nodules in the lungs are shown; melanoma nodules are observed as black tissue. Results are shown as the mean ± SEM (n = 3; ** *p* < 0.01).

**Figure 4 ijms-24-16947-f004:**
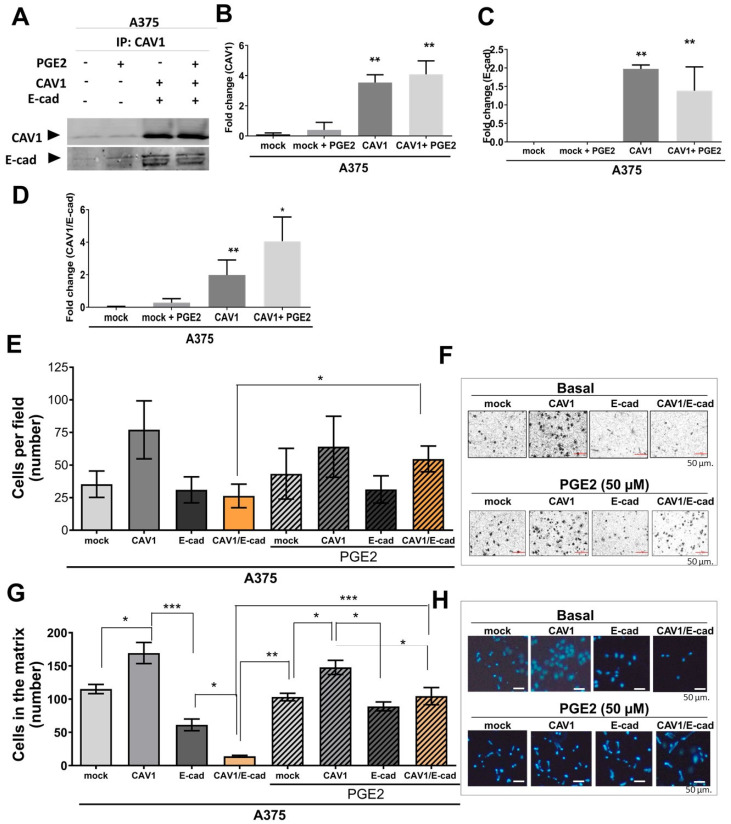
PGE2 increases migration and invasion of A375M (CAV1/E-cad) cells. A375 (Mock) and A375 (CAV1) cell lines cultured in the presence of IPTG were transiently transfected with pBATEM for 48 h to express E-cad. A375 (Mock), A375 (CAV1), A375 (E-cad), and A375 (CAV1/E-cad) cells were treated with 50 μM of PGE2 for 5 h. Then, the extracts were either prepared for immunoprecipitation experiments or cell migration and invasion measurements in transwell and matrigel assays, respectively. (**A**) CAV1 was immunoprecipitated from protein extracts using a polyclonal antibody. Immunoprecipitated CAV1 and co-immunoprecipitated E-cad were detected via Western blotting. (**B**) Representative Western blot of immunoprecipitates obtained in three independent experiments. E-cad and CAV1 protein levels were quantified in several experiments via densitometry and normalized to actin (mean ± SD, n = 3; ** *p* < 0.01). A375 (Mock), A375 (CAV1), A375 (E-cad), and A375 (E-cad/CAV1) cells were serum-deprived for 5 h and treated with PGE2 (50 μM) for 2 h. Then, CAV1 was immunoprecipitated from the protein extracts using a polyclonal antibody. Immunoprecipitated CAV1 and co-immunoprecipitated E-cad were detected via Western blotting. (**B**) Graph showing the CAV1 as normalized pixels (mean ± SEM) for each condition; n = 3; and ** *p* < 0.01 with respect to A375 (mock). (**C**) Graph showing E-cad as normalized pixels (mean ± SEM) for each condition; n = 3; ** *p* < 0.01 with respect to A375 (mock). (**D**) CAV1/E-cad as normalized pixels (mean ± SEM) for each condition; n = 3; and * *p* < 0.05 or ** *p* < 0.01 with respect to A375 (mock). (**E**) A375 (Mock), A375 (CAV1), A375 (E-cad), and A375 (E-cad/CAV1) cells were treated with 50 μM of PGE2 for 5 h. Then, 150.000 cells were seeded into Boyden chambers (transwells) in the presence of 50 μM PGE2. Migration was measured after 2 h. Significant differences are shown (* *p* < 0.05; n = 3). (**F**) Panels showing representative images of results obtained for cells migrating either without treatment (basal condition, **upper panel**) or when treated with 50 μM of PGE2 for 5 h (**lower panel**), in both cases stained with crystal violet. (**H**) Cells cultured in the presence or absence of 50 µM of PGE2 for 5 h were used for the invasion assays. Cells (150.000) were seeded into matrigel Boyden chambers in the presence of 50 μM of PGE2. Invasion was measured after 48 h. Graphs show averages (mean ± SEM; n = 3). Significant differences are indicated (* *p* < 0.05). (**G**) Panel showing representative images obtained in invasion assays for cells cultured in the absence (basal condition, **upper panel**) or presence (**lower panel**) of 50 μM of PGE2, in both cases stained with DAPI Bar = 50 μm. (* *p* < 0.05; ** *p* < 0.01; *** *p* < 0.001; n = 3).

**Figure 5 ijms-24-16947-f005:**
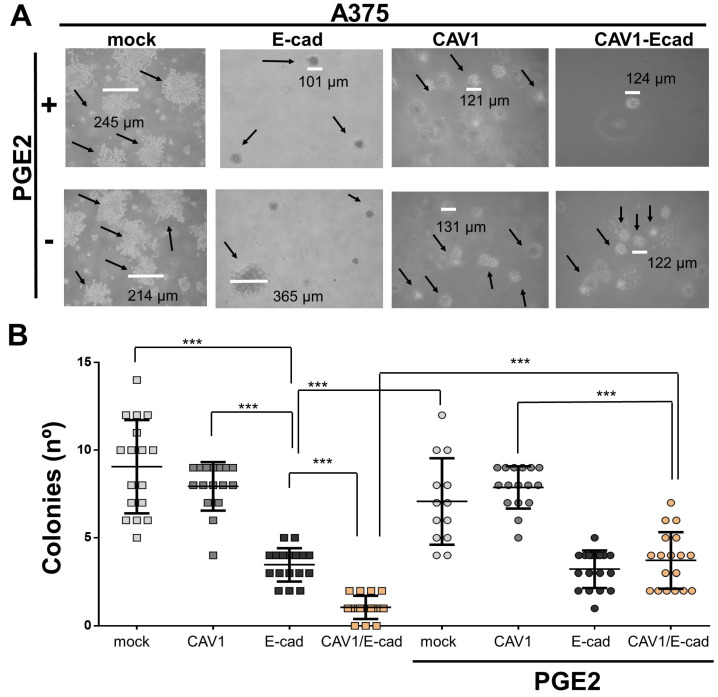
PGE2 increases colony formation by A375 (CAV1/E-cad) cells. (**A**) For the clonogenic assays, A375 (mock), A375 (CAV1), A375 (E-cad), and A375 (E-cadCAV1/E-cad) cells were previously treated for 5 h with 50 μM of PGE2. Cells (3 × 106) were mixed with 0.2% agarose and plated on low-adhesion plates for 24 h. Subsequently, cells were treated with PGE2 (diluted in medium supplemented with 10% FBS) for 14 days. Representative images of A375 cells treated (**upper panel**) or not treated (**lower panel**) with PGE2 are shown. The size of cell clusters is indicated in each image, in black arrows showed the clones, and in white line is the measured representative of clone. Colonies were then counted, but only colonies larger than 100 μm were considered. (**B**) Graph showing the number of cell colonies formed by A375 (mock), A375 (CAV1), A375 (E-cad), and A375 (CAV1/E-cad) cells treated or not with 50 μM PGE2. Results shown are the mean ± SEM (n = 3; *** *p* < 0.001).

**Figure 6 ijms-24-16947-f006:**
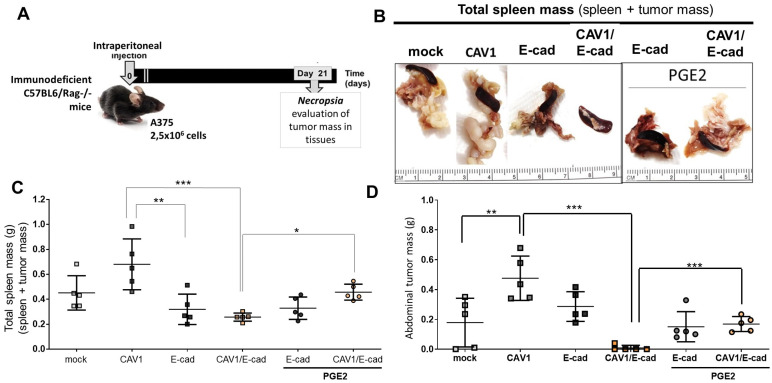
PGE2 enhances spleen and abdominal metastasis of A375 (CAV1/E-cad) cells. (**A**) Scheme showing the time course of the intraperitoneal carcinomatosis assays. Pathogen-free immunodeficient B16Rag1−/− mice between 5 and 7 weeks of age were used for the metastasis assays. Cells in culture were harvested, re-suspended in physiological serum, and pre-treated or not with 50 μM of PGE2 for 5 h. Then, 2.5 × 10^6^ A375 (mock), A375 (CAV1), A375 (E-cad), or A375 (CAV1/E-cad) cells were injected into the peritoneal cavity of the mice (day 0). After 21 days, all animals were necropsied. The total spleen mass was visualized and quantified. (**B**) Representative images of the total spleen mass (which includes the spleen and tissue-adherent tumor mass). (**C**) The graph shows the average total mass of spleens (g) in mice (mean ± SEM; n = 5). (**D**) The solid tumor mass in the abdominal cavity was also quantified. The graph shows the average abdominal tumor mass (g) in animals (mean ± SEM) from five mice (* *p* < 0.05, ** *p* < 0.01, and *** *p* < 0.001).

**Figure 7 ijms-24-16947-f007:**
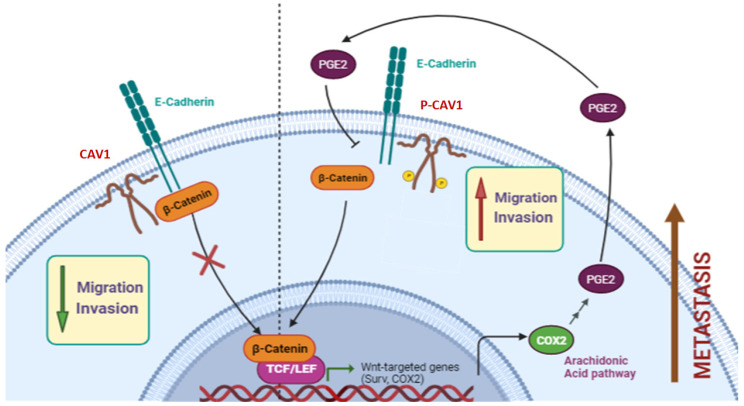
Summary model illustrating the effect of PGE2 on the CAV1/E-cad complex.

## Data Availability

Not applicable.

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
