# Peer review of "Prostaglandin E2 Exposure Disrupts E-Cadherin/Caveolin-1-Mediated Tumor Suppression to Favor Caveolin-1-Enhanced Migration, Invasion, and Metastasis in Melanoma Models"

_ijms, 2023, doi:10.3390/ijms242316947_

Round 1

Reviewer 1 Report

Comments and Suggestions for Authors

The work is interesting, and the experimental design overall is well structured. however, often the results reported in the text do not correspond to the figures indicated, for example in line 108-110 the Cav-1 levels of figure 1A and 1B are indicated while in fig. it should be 1C. Regarding immunoprecipitation, Fig. 1D is not absolutely clear. For the wb analysis monoclonal antibodies are used but in line 541 the secondary goat anti rabbit is indicated, why? For in vivo experiments the number of groups of animals used is not indicated, furthermore the NOD/SCID are indicated in line 567 while the C57BL/6 are shown in the figure. The growth graph of the tumor volume is not absolutely clear, furthermore it is not explained why the cell administrations are carried out in three different ways. I recognize that the importance of the work however absolutely needs to be reviewed.

Author Response

 Point by point response to reviewer comments

Comment: The work is interesting, and the experimental design overall is well structured.

Response: thank you for your positive assessment of our study.

Comment: however, often the results reported in the text do not correspond to the figures indicated, for example in line 108-110 the Cav-1 levels of figure 1A and 1B are indicated while in fig. it should be 1C.

Response:  Considering the reviewer´s comment, we modified the text in such way that it should now be easier to understand the figure showing the quantification values:

“Indeed, CAV1 levels increased in B16F10 (CAV1) cells in the presence of IPTG (Figs. 1A and 1B) around 10-fold, compared to B16F10 (mock) cells (18 ± 2.0 and 1.7 ± 0.3, respectively; Fig. 1B). E-cad was transiently expressed in B16F10 (mock) and B16F10 (CAV1) cells by transfection with pBATEM, a plasmid that encodes E-cad  (8.8 ± 1.0 and 7.3 ± 0.9, respectively, Fig. 1C). Following transfection, E-cad expression increased at least 7-fold in the B16F10 (E-cad) and B16F10 (CAV1/E-cad) cells, when compared to non-transfected cells (Figs. 1A and 1C)”.

Comment: Regarding immunoprecipitation, Fig. 1D is not absolutely clear.

Response: It not clear what precisely the reviewer means with this comment, but we reanalyzed the data and increased the contrast of the image to make things clearer. The cells B16F10 (mock) or B16F10 (CAV1) were transfected with the pBATEM plasmid and we  then performed the experiments with B16F10 (mock/E-cad) and B16F10 (CAV1/E-cad) cells. These cells were serum deprived for 5 h and treated  or not with PGE2 (20 μM) for 2 h. Then, CAV1 was immunoprecipitated from protein extracts using a polyclonal antibody. Immunoprecipitated CAV1 and co-immunoprecipitated E-cad were detected by immunoblotting. In the legend we include the “or not” to clarify the fact that some cells were treated with PGE2 and others were not.

Comment: For the wb analysis monoclonal antibodies are used but in line 541 the secondary goat anti rabbit is indicated, why?

Response: The polyclonal anti caveolin-1 (BD Transduction Laboratories) was included in methodology and was used for the CAV1 immunoprecipitation.

Comment: For in vivo experiments the number of groups of animals used is not indicated, furthermore the NOD/SCID are indicated in line 567 while the C57BL/6 are shown in the figure.

Response: Thanks for the comment. In the case of the C57BL/6 mice injected with different B16F10 cells, we mentioned in detail the number of mice used in the methodology section.

“B16F10 (Mock)(n=8/8), B16F10 (CAV1) (n=8/8), B16F10 (E-cad) (n=9/9), and B16F10 (CAV1/E-cad) (n=10/8) cells (1 x 105 cells), either pre-stimulated or not with PGE2 50 µM for 16 h (n=mice injected with pre-stimulated cells / mice injected with non-stimulated cells)”

Comment: The growth graph of the tumor volume is not absolutely clear, furthermore it is not explained why the cell administrations are carried out in three different ways.

Response: We agree that it is difficult to appreciate the changes in the graphs showing the tumor volumes as presented previously. Thus, we now separated the original data into three different graphs in order to clarify the messages that need to be conveyed:

B.1 Graph showing the time course experiment for mice injected with the cells without PGE2 pre-incubation, B.2 Time course for mice injected with cells that were pre-incubated with PGE2 and B.3 Graph showing  the curves for all time course experiments. The details concerning the number of mice employed are described in the methodology section.

Comment: I recognize that the importance of the work however absolutely needs to be reviewed.

Response: Once again thanks for your positive assessment of our study. As detailed above we have  addressed individually all your concerns.

Reviewer 2 Report

Comments and Suggestions for Authors

In this paper, Lorena Lobos-González and colleagues studied how prostaglandin E2 exposure disrupts E-cadherin/Caveolin-1 mediated tumor suppression to favor Caveolin-1-enhanced migration, invasion, and metastasis in melanoma models. This study is novel and well-presented; I have some minor concerns. 

It would be better to put some more references in the following sentences-

"Melanoma cells are characterized by the secretion of high levels of pro-inflammatory cytokines and prostaglandins (PG)."

"PGE2 is produced by a family of enzymes called cyclooxygenases, and in particular, PGE2 generated by cyclooxygenase-2 (COX-2) activity is associated with inflammatory processes and disease. For example, COX-2 is upregulated in a non-melanoma skin cancer (NMSC) model, which raises PGE2 levels." 

"More recent research has shown that melanoma cells proliferate [36] and migrate more when PGE2 receptor agonists are added, as opposed to when PGE2 receptor antagonists are present."

"Our earlier work identified CAV1 as a potent negative regulator of genes whose expression favors the development and progression of cancer."

In the method section, although the authors refer to the previous papers for the migration assay, a little description of this assay should also be explained in this text. 

In section 5.6, authors are recommended to mention how mice randomization and power calculation were done. 

In section 5.3, Does the blocking and primary antibody solution contain Tris? Please give all the catalog numbers of the primary and secondary antibodies.

The text should be checked for typos, grammar, and language correction throughout. 

Author Response

 Point by point response to reviewer comments

Comment: In this paper, Lorena Lobos-González and colleagues studied how prostaglandin E2 exposure disrupts E-cadherin/Caveolin-1 mediated tumor suppression to favor Caveolin-1-enhanced migration, invasion, and metastasis in melanoma models. This study is novel and well-presented; I have some minor concerns.

Response: we thank this reviewer for the positive assessment of our study.

Comment: It would be better to put some more references in the following sentences-"Melanoma cells are characterized by the secretion of high levels of pro-inflammatory cytokines and prostaglandins (PG)."

Response:

Thank you for your feedback. Indeed, several skin cells have been described to secrete high levels of pro-inflammatory cytokines and prostaglandins (PG). We have re-phrased this section as follows:

“In this regard, melanoma cells along with several types of skin cells (e.g. keratinocytes, sebocytes and, melanocytes) are characterized by the secretion of high levels of pro-inflammatory cytokines and prostaglandins (PG) in response to several pro-inflammatory stimuli [1–3].”Reviewed in:

  1. Neagu, M.; Constantin, C.; Caruntu, C.; Dumitru, C.; Surcel, M.; Zurac, S. Inflammation: A Key Process in Skin Tumorigenesis. Oncol. Lett. 2019, 17, 4068–4084, doi:10.3892/ol.2018.9735.
  2. Hölzel, M.; Tüting, T. Inflammation-Induced Plasticity in Melanoma Therapy and Metastasis. Trends Immunol. 2016, 37, 364–374, doi:10.1016/j.it.2016.03.009.
  3. Bald, T.; Quast, T.; Landsberg, J.; Rogava, M.; Glodde, N.; Lopez-Ramos, D.; Kohlmeyer, J.; Riesenberg, S.; van den Boorn-Konijnenberg, D.; Hömig-Hölzel, C.; et al. Ultraviolet-Radiation-Induced Inflammation Promotes Angiotropism and Metastasis in Melanoma. Nature 2014, 507, 109–113, doi:10.1038/nature13111.

Response: As requested, we have added the following references (lines 52-57):

PGE2 is produced by a family of enzymes called cyclooxygenases. Specifically cyclooxygenase-2 (COX-2) activity which is responsible for generating PGE2, has also been associated with inflammatory processes and disease [17]. For instance, COX-2 has been shown to be upregulated in a non-melanoma skin cancer (NMSC) model, where PGE2 levels have also been found to be increased. In this regard, increased expression of COX2 has also been observed in other types of cancer, such as colon cancer [17-18] which leads to increased levels of PGE2 involved in ROS production, inflammation, and disease progression [19–20].

For this section the following references 17 at 20 were added:

  1. Kim, Y.H.; Kim, K.J. Upregulation of Prostaglandin E2 by Inducible Microsomal Prostaglandin E Synthase-1 in Colon Cancer. Ann. Coloproctology 2022, 38, 153–159, doi:10.3393/ac.2021.00374.0053.
  2. Karpisheh, V.; Nikkhoo, A.; Hojjat-Farsangi, M.; Namdar, A.; Azizi, G.; Ghalamfarsa, G.; Sabz, G.; Yousefi, M.; Yousefi, B.; Jadidi-Niaragh, F. Prostaglandin E2 as a Potent Therapeutic Target for Treatment of Colon Cancer. Prostaglandins Other Lipid Mediat. 2019, 144, 106338, doi:10.1016/j.prostaglandins.2019.106338.
  3. Xue, X.; Shah, Y.M. Hypoxia-Inducible Factor-2α Is Essential in Activating the COX2/MPGES-1/PGE2 Signaling Axis in Colon Cancer. Carcinogenesis 2013, 34, 163–169, doi:10.1093/carcin/bgs313.
  4. J, S.; H, S.; Fb, Y.; B, L.; Y, Z.; Yt, Z. The Role of Cyclooxygenase-2 in Colorectal Cancer. Int. J. Med. Sci. 2020, 17, doi:10.7150/ijms.44439.

On the other hand, the following references were included in the discussion in the perspective of providing stronger evidence for the role of PGE2 in vivo:

  1. Smuđ Orehovec S, Dujmović A, Mijatović D, Mance M, Šarčević B. Immunohistochemical Expression of Matrix Metalloproteinase-1 and Cyclooxygenase-2 in Cutaneous Squamous Cell and Basal Cell Carcinoma. Acta Dermatovenerol Croat. 2021 Apr;291(1):8-20. PMID: 34477058.
  2. Kiraly AJ, Soliman E, Jenkins A, Van Dross RT. Apigenin inhibits COX-2, PGE2, and EP1 and also initiates terminal differentiation in the epidermis of tumor bearing mice. Prostaglandins Leukot Essent Fatty Acids. 2016 Jan;104:44-53. doi: 10.1016/j.plefa.2015.11.006. Epub 2015 Dec 21. PMID: 26802941.
  3. Rundhaug JE, Simper MS, Surh I, Fischer SM. The role of the EP receptors for prostaglandin E2 in skin and skin cancer. Cancer Metastasis Rev. 2011 Dec;30(3-4):465-80. doi: 10.1007/s10555-011-9317-9. PMID: 22012553; PMCID: PMC3236828.
  4. Inada M, Takita M, Yokoyama S, Watanabe K, Tominari T, Matsumoto C, Hirata M, Maru Y, Maruyama T, Sugimoto Y, Narumiya S, Uematsu S, Akira S, Murphy G, Nagase H, Miyaura C. Direct Melanoma Cell Contact Induces Stromal Cell Autocrine Prostaglandin E2-EP4 Receptor Signaling That Drives Tumor Growth, Angiogenesis, and Metastasis. J Biol Chem. 2015 Dec 11;290(50):29781-93. doi: 10.1074/jbc.M115.669481. Epub 2015 Oct 16. PMID: 26475855; PMCID: PMC4706013.

Response: As requested, we have added the reference 37 to the text.

"More recent research has shown that melanoma cells proliferate [36] and migrate more when PGE2 receptor agonists are added [37], as opposed to when PGE2 receptor antagonists are present [37]."

Vaid M, Singh T, Prasad R, Kappes JC, Katiyar SK. Therapeutic intervention of proanthocyanidins on the migration capacity of melanoma cells is mediated through PGE2 receptors and β-catenin signaling molecules. Am J Cancer Res. 2015 Oct 15;5(11):3325-38. PMID: 26807314; PMCID: PMC4697680

Response: As requested, we have also added the following references 39 and 44.

"Our earlier work identified CAV1 as a potent negative regulator of genes whose expression favors the development and progression of cancer [39.44]"

The reference was included: Reviewed in Lobos-Gonzalez et al., 2011 and Torres et al., 2007.

Comment: In the method section, although the authors refer to the previous papers for the migration assay, a little description of this assay should also be explained in this text. 

Response: As requested the assay has been described in greater detail. See section 5.1 in lines 554-559.

 “Briefly, the inserts were coated on the lower surface with 2 µg/ml fibronectin as a chemoattractant. Cells (1.5×105) re-suspended in serum-free medium were added to the upper chambers and serum-free medium was added to the bottom chambers. After 16 h, inserts were removed, washed and cells that migrated to the lower surface of the inserts were stained with 0.1% crystal violet in 2% ethanol and counted in an inverted microscope”

Comment: In section 5.6, authors are recommended to mention how mice randomization and power calculation were done. 

Response: We have now indicated in section 4.6 how the power calculation was done (80%). In lines 578-584  the details are provided.

“To calculate the minimum sample size, the r package, pwr2, was used. The “ss.1way” test was used for a one-way balanced ANOVA model (k: number of groups, alpha: level of significance, probability of type I error, beta: probability of type II error (power = 1 – beta), f is the effect size, delta: the smaller difference between groups k, sigma: the standard deviation B: is the number of iterations; the default value is 100”.

We also mentioned the reference where we first used these calulations to determine the number of mice required in our vivo model

Lobos-González, L., Aguilar, L., Díaz, J., Díaz, N., Urra, H., Torres, V.A., Silva, V., Fitzpatrick, C., Lladser, A., Hoek, K.S., Leyton, L., Quest, AF. E-cadherin determines Caveolin-1 tumor suppression or metastasis enhancing function in melanoma cells. Pigment Cell Melanoma Res. 2013; 26(4):555-70

Comment: In section 5.3, Does the blocking and primary antibody solution contain Tris? Please give all the catalog numbers of the primary and secondary antibodies.

Response: The precise composition of the blocking buffer and the first antibody dilution buffer are indicated In the methods section 4.3.

As requested information has been added to the section 4.1 in lines 496-500.

“Rabbit polyclonal anti-CAV1 (ab18199 rabbit polyclonal), mouse monoclonal anti-CAV1(C13620), mouse monoclonal anti-pY14-CAV1 (C 611339) and mouse monoclonal anti E-cad (C61018) antibodies were obtained from BD Transduction Laboratories, NJ, USA; monoclonal anti-β-actin antibodies (sc47778) were from Santa Cruz Biotechnology, TX, USA.

Comment: The text should be checked for typos, grammar, and language correction throughout. 

Response: The text has been carefully revised again by a native English speaker